# The Differential Impact of SRC Expression on the Prognosis of Patients with Head and Neck Squamous Cell Carcinoma

**DOI:** 10.3390/cancers11111644

**Published:** 2019-10-25

**Authors:** Francisco Hermida-Prado, Rocío Granda-Díaz, Nagore del-Río-Ibisate, M. Ángeles Villaronga, Eva Allonca, Irati Garmendia, Luis M. Montuenga, René Rodríguez, Aitana Vallina, César Alvarez-Marcos, Juan P. Rodrigo, Juana M. García-Pedrero

**Affiliations:** 1Department of Otolaryngology, Hospital Universitario Central de Asturias and Instituto de Investigación Sanitaria del Principado de Asturias (ISPA), Instituto Universitario de Oncología del Principado de Asturias, University of Oviedo, 33011 Oviedo, Spain; franjhermida@gmail.com (F.H.-P.); rocigd281@gmail.com (R.G.-D.); nagoredelrio@gmail.com (N.d.-R.-I.); angelesvillaronga@gmail.com (M.Á.V.); ynkc1@hotmail.com (E.A.); renerg.finba@gmail.com (R.R.); caalvarez@uniovi.es (C.A.-M.); 2Ciber de Cáncer, CIBERONC, Instituto de Salud Carlos III, Av. Monforte de Lemos, 3-5, 28029 Madrid, Spain; lmontuenga@unav.es; 3Program in Solid Tumors, Center for Applied Medical Research (CIMA); Department of Pathology, Anatomy and Physiology, University of Navarra, and Navarra’s Health Research Institute (IDISNA), 31008 Pamplona, Spain; igarmendia@alumni.unav.es; 4Department of Pathology, Hospital Universitario Central de Asturias and Instituto de Investigación Sanitaria del Principado de Asturias (ISPA), Instituto Universitario de Oncología del Principado de Asturias, University of Oviedo, 33011 Oviedo, Spain; alaicla@hotmail.es

**Keywords:** head and neck squamous cell carcinoma, immunohistochemistry, SRC, prognosis, larynx, pharynx

## Abstract

Aberrant SRC expression and activation is frequently detected in multiple cancers, and hence, targeting SRC has emerged as a promising therapeutic strategy. Different SRC inhibitors have demonstrated potent anti-tumor activity in preclinical models, although they largely lack clinical efficacy as monotherapy in late-stage solid tumors, including head and neck squamous cell carcinomas (HNSCC). Adequate selection and stratification of patients who may respond to and benefit from anti-SRC therapies is therefore needed to guide clinical trials and treatment efficacy. This study investigates the prognostic significance of active SRC expression in a homogeneous cohort of 122 human papillomavirus (HPV)-negative, surgically treated HNSCC patients. Immunohistochemical evaluation of the active form of SRC by means of anti-SRC Clone 28 monoclonal antibody was specifically performed and subsequently correlated with clinical data. The expression of p-SRC (Tyr419), total SRC, and downstream SRC effectors was also analyzed. Our results uncovered striking differences in the prognostic relevance of SRC expression in HNSCC patients depending on the tumor site. Active SRC expression was found to significantly associate with advanced disease stages, presence of lymph node metastasis, and tumor recurrences in patients with laryngeal tumors, but not in the pharyngeal subgroup. Multivariate Cox analysis further revealed active SRC expression as an independent predictor of cancer-specific mortality in patients with laryngeal carcinomas. Concordantly, expression of p-SRC (Tyr419) and the SRC substrates focal adhesion kinase (FAK) and the Arf GTPase-activating protein ASAP1 also showed specific associations with poor prognosis in the larynx. These findings could have important implications in ongoing Src family kinase (SFK)-based clinical trials, as these new criteria could help to improve patient selection and develop biomarker-stratified trials.

## 1. Introduction

Head and neck squamous cell carcinoma (HNSCC) represents the sixth most common cancer worldwide. The latest advancements in cancer diagnosis and treatment have led to only modest improvements of survival rates for HNSCC patients [1]. It has become clear by omics studies that HNSCC is a highly complex and heterogeneous disease [2], involving multiple different genetic and molecular alterations that ultimately hamper our ability to accurately predict aggressive tumor behavior. The identification of novel markers capable of distinguishing the biological behavior of tumors could certainly contribute to improve predictability beyond the current clinicopathological markers.

SRC belongs to the highly conserved family of non-receptor protein tyrosine kinases known as the Src family kinases (SFKs) that includes SRC, BLK, FGR, FRK, FYN, HCK, LCK, LYN, YES, and YRK [3,4]. Each family member has shown a different expression pattern and tissue distribution. SRC is one of the oldest and most investigated proto-oncogenes. Mounting evidence demonstrates that SRC plays a pivotal role at different stages of tumorigenesis [4,5,6], thereby modulating multiple oncogenic signaling pathways and biological processes essential for the malignant phenotype, such as proliferation, cell adhesion, motility, invasion, and angiogenesis [5,6,7,8,9]. It has been reported that SRC interacts with multiple receptor tyrosine kinases (RTKs), including, among others, epidermal growth factor receptors (EGFR and HER2), insulin growth factor receptor IGF-1R, fibroblast growth factor receptor, platelet-derived growth factor receptor, hepatocyte growth factor receptor c-MET, and c-Kit [10]. SRC is a critical player in the regulation of cellular adhesion mediated by integrins, epithelial-to-mesenchymal transition via E-cadherin suppression, and focal adhesions by activation of focal adhesion kinase (FAK) [11,12,13]. In addition, SRC regulates the EGF-dependent actin cytoskeleton reorganization via p190 phosphorylation [14]. Cortactin (CTTN) is another important SRC substrate that participates in actin remodeling and invadopodium formation. It has been described that SRC-mediated phosphorylation of CTTN modulates its interaction with FAK at focal adhesions, thereby promoting cell motility. On this basis, it has been suggested that CTTN may act as a bridge between actin filaments and focal adhesions [15]. The Arf GTPase-activating protein (GAP) ASAP1 (also known as AMAP1 or DDEF1) has also emerged as a key SRC substrate that regulates cytoskeleton dynamics and invadopodium formation by binding to both CTTN and paxillin, thereby playing a critical role in promoting invasion and metastasis [16,17].

Elevated SRC expression and activity has been frequently detected in multiple cancers, including breast, ovarian, colon, pancreatic, gastric, hepatocellular, lung, bladder esophageal, and HNSCC [9,18,19,20,21]. These observations led to rapid development of SRC inhibitors, such as dasatinib (BMS354825), saracatinib (AZD0530), and bosutinib (SKI-606), which have been actively tested in preclinical settings and also in clinical trials for multiple cancers [4,21,22,23,24,25,26,27,28,29,30,31,32]. However, disappointingly, SRC inhibitors did not show any significant activity as monotherapeutic agents in the treatment of patients with advanced-stage solid tumors [21,23,24,25,29,30], including HNSCC [33,34].

Human SRC protein is a tyrosine kinase structurally composed of four Src-homology (SH1-4) domains that mediate the regulation of SRC kinase activity and also protein interaction with a large number of substrates to form intracellular signaling complexes [5,7,10]. In addition, full activation of SRC is tightly regulated and dependent on the phosphorylation of residue Tyr419, while Tyr530 phosphorylation is involved in its inactivation. The generation of a monoclonal antibody (Clone 28) specific for the C-terminal regulatory domain of human c-Src that selectively recognizes the active form of SRC (not phosphorylated at Tyr530) was previously reported [35]. Hence, this antibody emerged as a useful tool to specifically detect the active form of SRC protein in both tissues and cells.

The present study is the first to investigate the expression pattern and clinical significance of active SRC in a homogeneous cohort of 122 human papillomavirus (HPV)-negative HNSCC patients treated by surgery at the same institution. Our results reveal clear differences in the prognostic relevance of SRC expression in HNSCC patients depending on the tumor site. Thus, aberrant SRC expression/activation was found to significantly associate with advanced disease stages, presence of lymph node metastasis, and tumor recurrences in patients with laryngeal tumors, but not in the pharyngeal subgroup. Moreover, active SRC expression now emerges as an independent predictor of cancer-specific mortality in laryngeal cancer. Similarly, the expression of p-SRC (Tyr419) and various downstream SRC effectors such as FAK and ASAP1 significantly and consistently correlated with active SRC expression and also specifically predicted poor prognosis in the larynx. These novel findings could be useful to improve the selection of patients who can benefit from treatment with anti-SRC therapies. Accordingly, biomarker-based patient stratification could help to refine eligibility criteria and guide clinical trials, and to ultimately improve treatment efficacy and long-term clinical outcomes.

## 2. Materials and Methods

### 2.1. Patients and Tissue Specimens

Surgical tissue specimens were collected from 122 patients with laryngeal or hypopharyngeal squamous cell carcinoma surgically treated at the Hospital Universitario Central de Asturias (HUCA) between 1996 and 2005, in accordance with approved Institutional Review Board guidelines. Experimental procedures were performed in accordance with the Declaration of Helsinki. Written informed consent was obtained from all patients. Formalin-fixed paraffin-embedded (FFPE) tissue samples and data from donors were provided by the Principado de Asturias BioBank (PT17/0015/0023), integrated in the Spanish National Biobanks Network, and histological diagnosis was confirmed by an experienced pathologist. Samples were processed following standard operating procedures with the appropriate approval of the Ethical and Scientific Committees of the HUCA and the Regional CEIC from Principado de Asturias for the project PI16/00280 (approval number: 70/16; date: 5 May 2016).

All patients had a single primary tumor and microscopically clear surgical margins. Patients did not receive any treatment prior to surgery. Only five patients were women. The mean age was 60 years (range 38 to 86 years). Of the 122 patients, 119 were habitual tobacco smokers, 71 moderate (1–50 pack-years) and 48 heavy (>50 pack-years), and 107 were habitual alcohol drinkers; further, 66 (54%) of the 122 patients received postoperative radiotherapy. The characteristics of the patients studied and the clinicopathologic features of their tumors are summarized in Table 1 and Appendix A. The stage of disease was determined after the surgical resection of the tumor according to the TNM system of the International Union against Cancer (7th Edition). The histological grade was determined according to the degree of differentiation of the tumor (Broders’ classification). The HPV status was available for all the patients, determined as previously reported [36].

### 2.2. Immunohistochemical Analysis of Patient Samples

Three morphologically representative areas were selected from each individual tumor block to construct five tissue microarray (TMA) blocks, as described previously [36]. Each TMA also contained three cores of normal epithelium as an internal control. TMA blocks were cut into 3 μm sections and dried on Flex IHC microscope slides (Dako, Glostrup, Denmark). TMA sections were deparaffinized, and antigen retrieval was carried out by using Envision Flex Target Retrieval solution (Dako, Glostrup, Denmark), high pH or low pH (for active SRC) or proteinase K (for ASAP1). Staining was performed at room temperature on an automatic staining workstation (Dako Autostainer Plus) with SRC (active) monoclonal antibody Clone 28 (Thermo Fisher Scientific #AHO0051) at 1:300 dilution, phospho-SRC (Y419) antibody (Invitrogen # 44-660G) at 1:80 dilution, total SRC (Santa Cruz Biotechnology # sc-8056) at 1:100, anti-ASAP1 antibody (abcam # 11011) at 1:750, mouse anti-cortactin monoclonal antibody Clone 30 (BD Biosciences Pharmingen # 610049) at 1:200 dilution, or mouse anti-FAK monoclonal antibody Clone 4.47 (Merck Millipore # 05-537) at 1:250 dilution using the Dako EnVision Flex+ Visualization System (Dako Autostainer, Denmark). Counterstaining with hematoxylin was the final step.

Immunostaining was scored blinded to clinical data by two independent observers. Since SRC, ASAP1, CTTN, and FAK staining showed a homogeneous distribution, a semiquantitative scoring system was applied, and staining intensity was scored as negative (0), weakly (1+), moderately (2+), or strongly positive (3+). Scores of ≥2 were considered as positive expression. p-SRC (Tyr419) expression was detected in both the nucleus and cytoplasm of tumor cells and scored as negative (0) versus positive expression.

### 2.3. Statistical Analyses

Statistical analyses were performed using the SPSS 15.0 software package (SPSS Inc., Chicago, IL, USA). The χ^2^ and Fisher’s exact tests were used for comparison between categorical variables. Cox proportional hazards models were used for univariate and multivariate analyses. The hazard ratios (HR), 95% confidence intervals (CI), and *p* values are reported. Kaplan–Meier survival curves were also plotted. Differences between survival times were analyzed by the log-rank method. All tests were two-sided.

## 3. Results

### 3.1. Detection of Active SRC in HNSCC Tissue Specimens

Immunohistochemical analysis of active SRC expression was performed on tissue specimens from 122 HNSCC patients. Immunostaining was successfully evaluated in 116 (95%) of the 122 cases. Of these 116 tumors, 88 (76%) exhibited positive SRC expression preferentially detected in the cytoplasm, although some cases also displayed protein enrichment at the cell membrane (Figure 1A–C). Normal epithelium showed positive staining in the basal cell layer and negligible expression in the most differentiated layers (Figure 1D). In addition, immunohistochemical analysis of p-SRC (Tyr419) and total SRC was also performed and correlated with active SRC expression (not phosphorylated at Tyr530). Positive p-SRC (Tyr419) staining was mainly detected in the nucleus in 98 (88%) of the tumors, and some cases also exhibited cytoplasmic staining (11 tumors) (Figure 1E–H). A significant positive correlation was observed between active SRC expression and nuclear p-SRC (Tyr419) (Spearman correlation coefficient 0.305, *p* = 0.001) but not cytoplasmic p-SRC (Tyr419) (Spearman correlation coefficient −0.023, *p* = 0.810). Concordantly, total SRC expression also exhibited both nuclear and cytoplasmic patterns (Figure 1I–K) and showed a significant correlation with active SRC expression (Spearman correlation coefficient 0.318, *p* = 0.001). The expression of p-SRC (Tyr419) was also confirmed by Western blot analysis in a subset of tumor samples compared to patient-matched normal tissues and in a panel of HNSCC-derived cell lines (Appendix A). Consistent with the IHC data, p-SRC (Tyr419) levels were increased in tumors compared to patient-matched normal epithelia (Patients 1–3) as well as HNSCC cells. Total SRC expression levels were also found to increase in tumors compared to the normal counterparts (Patients 1,2, and 4).

### 3.2. Correlations with Clinicopathological Parameters and Disease Outcome

We next assessed the associations between expression of active SRC and the clinicopathological parameters and disease outcome. As shown in Table 1, positive SRC expression was significantly associated with the presence of lymph node metastasis (*p* = 0.020), and advanced disease stage (*p* = 0.048). In addition, SRC expression was strongly and significantly associated with tumor recurrence (*p* = 0.002). Accordingly, patients carrying SRC-positive tumors showed a significantly higher rate of cancer-related deaths (*p* = 0.001) (Table 1).

### 3.3. Differential Clinical Significance of Active SRC Expression Depending on the Tumor Location

When the clinical relevance of active SRC was analyzed separately by tumor site in our cohort of HNSCC patients, striking differences were observed (Table 2). Positive SRC expression was more frequent in bigger tumor sizes (T3, T4) and cases with lymph node metastasis (N+) in both locations, although the differences were not statistically significant. It is noteworthy, however, that active SRC expression was strongly and significantly correlated with tumor recurrences and tumor-associated deaths specifically in the larynx (*p* < 0.001) but not the pharynx (*p* = 1.00).

### 3.4. Impact of SRC Expression on Patients’ Survival

Univariate survival analysis showed a significant correlation between SRC expression and reduced disease-specific survival in the total cohort of HNSCC patients (log-rank test, *p* = 0.002; Figure 2A). Interestingly, clear differences were observed when we evaluated the impact of SRC expression on patient survival in the different tumor locations. Thus, active SRC expression was significantly correlated with a shorter disease-specific survival in the larynx (log-rank test, *p* < 0.001) but not in the pharynx (log-rank test, *p* = 0.940; Figure 2B,C). Multivariate Cox analysis including pT classification (dichotomized as T1,T2 versus T3,T4), pN classification (dichotomized as N0 versus N+), and active SRC expression further demonstrated that active SRC expression (HR = 12.78; 95% CI 1.7–96.11; *p* = 0.013) and cervical lymph node metastasis (HR = 3.60; 95% CI 1.39–9.285; *p* = 0.008) were potent independent predictors of shorter disease-specific survival in patients with laryngeal tumors.

Similarly, nuclear p-SRC (Tyr419) was also significantly associated with shorter disease-specific survival (log-rank test, *p* = 0.043; Figure 2D) with marked differences depending on the tumor site (Figure 2E,F). Furthermore, the possible impact of full SRC activation on patient prognosis was assessed by analyzing simultaneously active SRC expression and p-SRC (Tyr419). Interestingly, we found that patients harboring both active SRC expression and p-SRC (Tyr419) (Group 2) exhibited significantly shorter survival than did patients with positive expression of either active SRC or p-SRC (Tyr419) (Group 1) or than did patients with negative expression of both active SRC expression and p-SRC (Tyr419) (Group 0) (log-rank test, *p* = 0.006; Figure 2G). Furthermore, these differences were specifically observed in the larynx (log-rank test, *p* = 0.003; Figure 2H), but not the pharynx (log-rank test, *p* = 0.885; Figure 2I).

### 3.5. Correlations between the Expression of Active SRC and SRC-Related Proteins

We next evaluated by immunohistochemistry the expression of various SRC effectors, such as CTTN, FAK, and ASAP1, known to regulate tumor invasion, metastasis, and aggressive phenotypes in HNSCC and other cancers [11,12,13,14,15,16,17,37,38,39]. Positive CTTN, FAK, and ASAP1 expression was respectively detected in 59 (51%), 83 (72%), and 64 (55%) tumors in our cohort of HNSCC patients, predominantly cytoplasmic and with membrane enrichment in some cases (Figure 3). The expression levels of FAK and ASAP1 were found to significantly correlate with positive SRC expression (*p* = 0.015 and *p* = 0.017, respectively; Table 3). The potassium channel HERG1 was previously reported to form complexes with FAK [40], and is associated with aggressive tumor behavior and poor prognosis in HNSCC [41]. The expression of HERG1 was also found to significantly correlate with SRC expression in our HNSCC cohort (*p* = 0.011, Table 3).

When examining separately each tumor site subgroup (Table 4), we strikingly observed that the correlations between SRC expression and the SRC-related proteins FAK, ASAP1, and HERG1 were specific to the larynx and not the pharynx. Consistent with these results, the expression of the SRC substrates FAK, CTTN, and ASAP1 also showed a distinct impact on patient survival depending on the tumor site (Figure 4).

## 4. Discussion

Aberrant SRC expression and activation is frequent in a wide variety of cancers and has been identified as a central node in numerous oncogenic pathways, thereby playing critical roles in tumor formation, progression, and dissemination [4,5,6,7,8,9]. Hence, targeting SRC has emerged as a promising therapeutic strategy for cancer treatment, and a number of SRC inhibitors have subsequently been developed and tested [4,21,22,23,24,25,26,27,28,29,30]. Overall, SRC inhibitors have demonstrated potent anti-tumor activity in preclinical models, although they are largely ineffective for the treatment of late-stage solid tumors [21,23,24,25,29,30]. In the specific context of HNSCC, dasatinib and saracatinib robustly inhibited cell proliferation, migration, and invasion in preclinical models [31,32,42,43]; however, these compounds did not show clinical efficacy as monotherapy in patients with advanced metastatic disease [33,34]. This therefore reflects the need for accurate response markers and adequate patient stratification to guide treatment with SRC inhibitors, which will undoubtedly contribute to improving clinical effectiveness and disease outcome.

It seems quite reasonable that those tumors harboring aberrant SRC activation and function are more likely to be effectively targeted by anti-SRC therapies. Indeed, SRC pathway activation has been positively correlated with sensitivity to treatment with dasatinib and saracatinib in different cancer types [44,45,46], suggesting that SRC activation could potentially serve as a biomarker to guide SRC targeting and clinical efficacy. The present study investigated the clinical significance of active SRC expression in HNSCC patients, by means of immunohistochemical detection with Clone 28 antibody specifically recognizing the active form of SRC [35,47]. In addition, the expression levels of p-SRC (Tyr419), total SRC, and various downstream SRC-related proteins were also evaluated by IHC and revealed significant correlations with active SRC expression in the total HNSCC cohort and the laryngeal subgroup. Notably, our results evidenced clear differences in the clinical impact of active SRC expression on patient survival depending on the tumor site. Active SRC specifically emerged as an independent predictor of cancer-specific mortality in patients with laryngeal tumors, but not in the pharyngeal subgroup.

Consistent with these findings, the expression of p-SRC (Tyr419) and the SRC substrates FAK and ASAP1 also showed specific associations with poor prognosis in the larynx. We also found that CTTN was another specific predictor of poor prognosis in the larynx but not in the pharynx, in agreement with our previous study [37]. However, a significant correlation between CTTN and SRC expression was not observed in our HNSCC group. Together these observations fit with the well-established oncogenic role of SRC and its downstream effectors favoring aggressive tumor phenotypes by promoting invasion and metastatic dissemination. Specifically, CTTN, FAK, and ASAP1 have been demonstrated to be key regulators of tumor invasion, metastasis, and aggressive phenotypes in HNSCC and other cancers [11,12,13,14,15,16,17,37,38,39]. In addition, we previously uncovered that the potassium channel HERG1 plays a fundamental role in early stages of HNSCC tumorigenesis and disease progression [41]. HERG1 expression was thus associated with aggressive tumor behavior and poor prognosis. This information was further and significantly extended in the present study, thereby uncovering a differential association of HERG1 expression with SRC expression depending on the HNSCC tumor site.

Similarly, SRC expression has been correlated with nodal metastasis, advanced clinical stages, recurrence, and poor prognosis in patients with oral carcinomas [48]. Increased p-SRC (Tyr419) levels have also been detected in nasopharyngeal carcinomas, both in tissue samples and in plasma, and correlated with tumor aggressiveness, distant metastasis, and unfavorable prognosis [49]. More controversially, SRC inhibitor monotherapy did not demonstrate any significant benefit in patients with metastatic HNSCC. These studies were performed in unselected cohorts of patients. Likewise, the lack of effectiveness could reflect the involvement of SRC in the first steps of the metastatic cascade to facilitate the migration and invasion of tumor cells, but not in the late stage when tumors are already disseminated.

In addition, we recently demonstrated that dasatinib and saracatinib enhanced cancer stem cell (CSC) properties in HNSCC models [50], which could also represent a plausible underlying reason to explain the lack of clinical efficacy as monotherapy in HNSCC patients. Remarkably, while dasatinib was unable to eliminate CSC subpopulations or to exhibit any significant anti-cancer activity in mouse xenografts, the mithramycin analog EC-8042 effectively targeted these deleterious effects and robustly diminished tumor growth in vivo. In line with these findings, it has been reported that dasatinib worsened the anti-neoplastic effects of cetuximab and radiation in mouse HNSCC models [51]. Moreover, saracatinib showed no effect on tumor growth but blocked perineural invasion and nodal metastasis in an orthotopic model of oral cancer [52]. Similarly, the EGFR inhibitor erlotinib, but not dasatinib, was found to significantly reduce tumor size in a randomized trial with operable HNSCC patients [53]. Together these data reinforce a major role for SRC favoring tumor invasion and metastasis rather than sustaining tumor growth, in good agreement with the presented results herein.

As an attempt to enhance the clinical benefit of SRC inhibitors, various combinational regimens with other anti-cancer agents have been designed and have shown promising results. Targeting SRC was effective in overcoming trastuzumab resistance and eliminating trastuzumab-resistant tumors in vivo [54]. SRC inhibition was also shown to overcome resistance to HER2 inhibitors, restoring lapatinib sensitivity [55], and to reduce tumor growth in Met-driven tumors [56]. Dual inhibition of SRC and MET led to synergistic cytotoxic effects in HNSCC models [57], and this combination targeting has also emerged as a promising therapeutic strategy for colon cancer [58]. In addition, combination treatment of dasatinib with EC-8042 in HNSCC models demonstrated favorable complementary anti-proliferative, anti-stemness, and anti-invasive effects [50], suggesting this novel combinational strategy for clinical testing in HNSCC patients. Nevertheless, it should be emphasized that adequate preselection of patients who may respond to and benefit from anti-SRC therapies is fundamental to guide clinical trials, as well as to identify reliable response biomarkers to improve stratification, treatment efficacy, and, ultimately, clinical outcome.

## 5. Conclusions

We herein provided unprecedented evidence for the differential clinical impact of SRC expression in HNSCC patients depending on the tumor site. Active SRC expression specifically emerged as an independent predictor of poor prognosis in patients with laryngeal cancer. Similarly, the expression levels of p-SRC (Tyr419) and various downstream SRC effectors such as FAK and ASAP1 significantly and consistently correlated with active SRC expression and specifically predicted poor prognosis in the larynx. Our findings could have important implications for various ongoing SFK-based clinical trials, as these new criteria could help to improve the selection and biomarker-based stratification of patients who may benefit from treatment with SRC inhibitors.

## Figures and Tables

**Figure 1 cancers-11-01644-f001:**
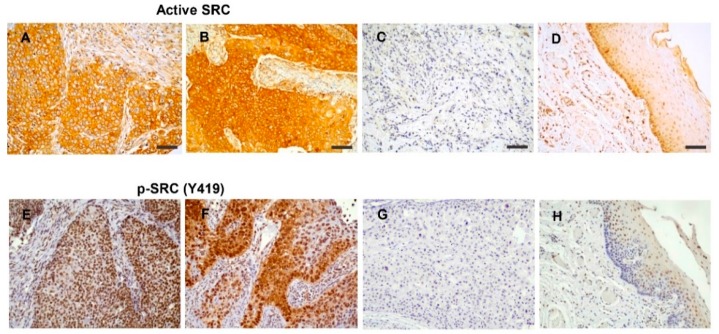
Immunohistochemical analysis of SRC expression in head and neck squamous cell carcinoma (HNSCC) tissue specimens. Representative examples of HNSCC showing positive active SRC staining (**A**,**B**, cytoplasmic and membrane enrichment), negative staining (**C**), and normal adjacent epithelia (**D**). Representative examples of tumors with positive p-SRC (Tyr419) staining (**E**, nuclear and **F**, cytoplasmic and nuclear), negative staining (**G**), and normal adjacent epithelia (**H**). Representative examples of tumors showing positive total SRC staining (**I**,**J**, cytoplasmic and nuclear) and negative staining (**K**). Magnification 20×. Scale bars = 50 μm.

**Figure 2 cancers-11-01644-f002:**
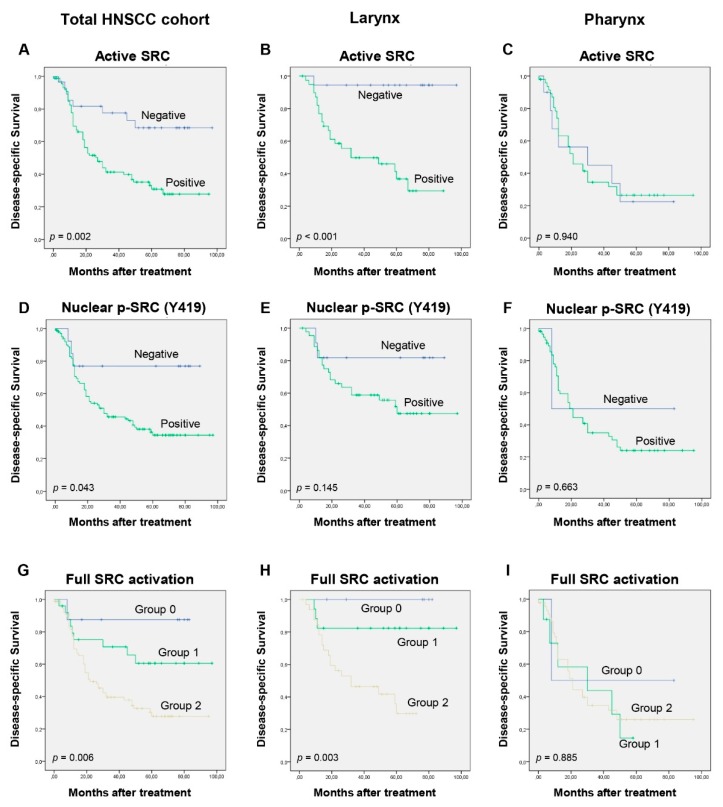
Impact of SRC expression on patient survival. Kaplan–Meier disease-specific survival curves categorized by active SRC expression (**A**–**C**) or nuclear p-SRC (Tyr419) (**D**–**F**) in the total cohort of HNSCC patients or in the laryngeal and pharyngeal subgroups. Kaplan–Meier disease-specific survival curves considering jointly both active SRC expression and nuclear p-SRC (Tyr419) dichotomized into three groups: positive expression of both active SRC expression and nuclear p-SRC (Tyr419) (Group 2), positive expression of either active SRC or nuclear p-SRC (Tyr419) (Group 1), or none (Group 0) (**G**–**I**) in the total HNSCC cohort and in the laryngeal and pharyngeal subgroups. *p* values were estimated using the log-rank test.

**Figure 3 cancers-11-01644-f003:**
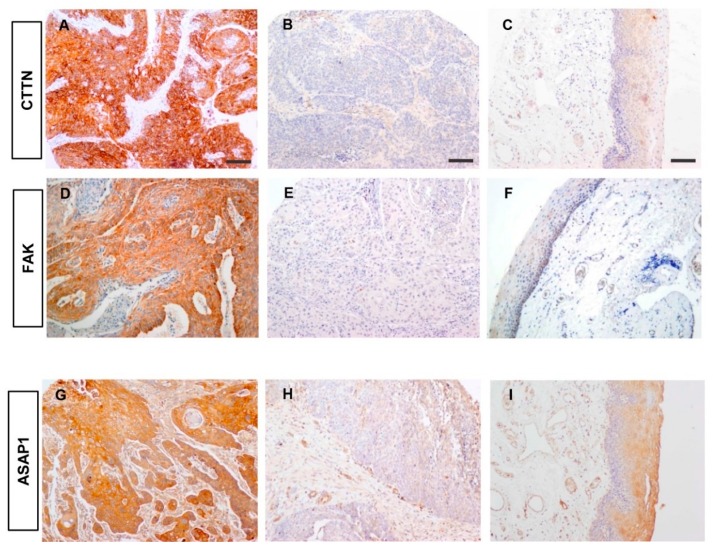
Immunohistochemical analysis of downstream SRC-related proteins in HNSCC tissue specimens. Representative examples of HNSCC showing strong positive cortactin (CTTN) staining (**A**), negative staining (**B**), and normal adjacent epithelia (**C**). Representative examples of tumors with positive focal adhesion kinase (FAK) staining (**D**), negative staining (**E**), and normal adjacent epithelia (**F**). Representative examples of tumors with positive staining for the Arf GTPase-activating protein ASAP1 (**G**), negative staining (**H**), and normal adjacent epithelia (**I**). Magnification 10×. Scale bars = 100 μm.

**Figure 4 cancers-11-01644-f004:**
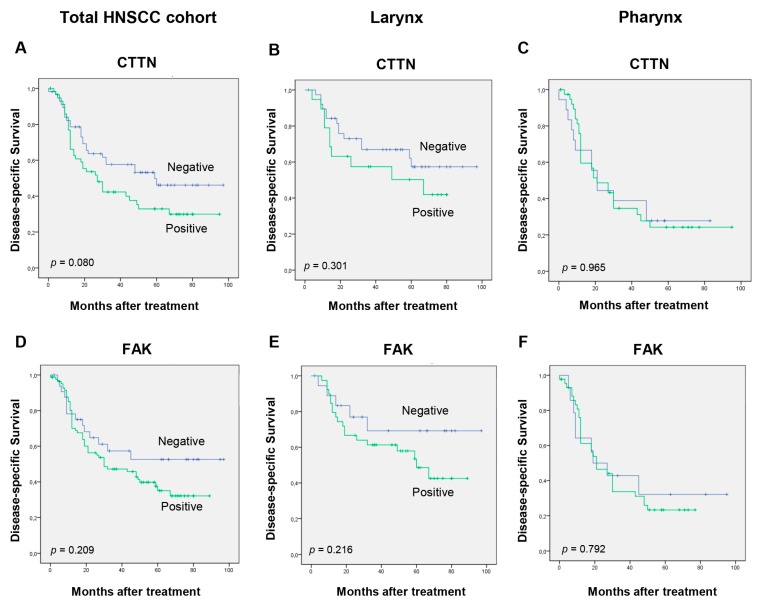
Impact of SRC-related proteins on patient survival. Kaplan–Meier disease-specific survival curves categorized by CTTN expression (**A**–**C**), FAK expression (**D**–**F**), and ASAP1 expression (**G**–**I**) in the total cohort of HNSCC patients or in the laryngeal and pharyngeal subgroups. *p* values were estimated using the log-rank test.

**Table 1 cancers-11-01644-t001:** Associations between the expression of active SRC and clinicopathological parameters, recurrence, and disease outcome.

Characteristic	No.	Active SRC Expression (%)	*P*
pT classification			
T1, T2	32	21 (66)	0.145 ^†^
T3, T4	84	67 (79)	
pN classification			
N0	39	24 (62)	0.020 ^†^
N1–3	77	64 (83)	
Disease stage			
I, II	15	8 (53)	0.048 ^†^
III, IV	101	80 (79)	
Pathological grade			
Well differentiated	33	22 (67)	0.286 ^#^
Moderately differentiated	53	41 (77)	
Poorly differentiated	30	25 (83)	
Site			
Pharynx	58	48 (83)	0.128 ^†^
Larynx	58	40 (69)	
Tumor Recurrence ^‡^			
No	26	14 (54)	0.002 ^†^
Yes	68	58 (85)	
Disease status			
Alive without disease	30	16 (53)	0.001 ^†^
Dead of index cancer	64	56 (88)	
___			
Died of other causes	22	16 (73)	
Total Cases	116	88 (76)	

^#^ Chi-square and ^†^ Fisher’s exact tests. ^‡^ The 22 patients who died of cancer-unrelated causes were excluded from the recurrence analysis.

**Table 2 cancers-11-01644-t002:** Associations of active SRC expression with clinical and follow-up data for patients with pharyngeal and laryngeal tumors.

Characteristic	Pharynx (No. = 58)			Larynx (No. = 58)		
	No.	Active SRC (%)	*P*	No.	Active SRC (%)	*p*
pT classification						
T1, T2	17	13 (76)		15	8 (53)	
T3, T4	41	35 (85)	0.458 ^†^	43	32 (74)	0.194 ^†^
pN classification						
N0	10	7 (70)		29	17 (59)	0.155 ^†^
N1–3	48	41 (85)	0.353 ^†^	29	23 (79)	
Disease stage						
I, II	5	4 (80)	1.000 ^†^	10	4 (40)	0.055 ^†^
III, IV	53	44 (83)		48	36 (75)	
Pathological grade						
Well differentiated	12	10 (83)		21	12 (57)	
Moderately differentiated	26	22 (85)	0.917 ^#^	27	19 (70)	0.177 ^#^
Poorly differentiated	20	16 (80)		10	9 (90)	
Tumor Recurrence ^‡^						
No	8	7 (88)	1.000 ^†^	18	7 (39)	<0.001 ^†^
Yes	42	34 (81)		26	24 (92)	
Disease status						
Alive without disease	10	8 (80)	1.000 ^†^	20	8 (40)	<0.001 ^†^
Dead of index cancer	40	33 (83)		24	23 (96)	
___						
Died of other causes	8	7 (88)		14	9 (64)	

^#^ Chi-square and ^†^ Fisher’s exact tests. ^‡^ The 8 patients (pharynx group) and 14 patients (larynx group) who died from causes not related to the index tumor were excluded from the recurrence analysis.

**Table 3 cancers-11-01644-t003:** Associations between the expression of active SRC and downstream SRC-related proteins.

Molecular Feature	No.	Active SRC (%)	*p* ^#^
CTTN protein expression			(0.050)
Negative	57	42 (74)	0.594
Positive (scores 2–3)	59	46 (78)	
FAK protein expression			(0.225)
Negative	33	20 (61)	0.015
Positive (scores 2–3)	83	68 (82)	
ASAP1 protein expression			(0.221)
Negative	52	34 (65)	0.017
Positive (scores 2–3)	64	54 (84)	
HERG1 protein expression			(0.240)
Negative	14	7 (50)	0.011
Positive (scores 2–3)	98	79 (81)	

^#^ Spearman correlation coefficient (in parentheses) with the associated *p* value.

**Table 4 cancers-11-01644-t004:** Associations between active SRC expression and SRC-related proteins by tumor site.

Molecular Feature	Pharynx (No. = 58)			Larynx (No. = 58)		
	No.	Active SRC (%)	*p* ^#^	No.	Active SRC (%)	*p* ^#^
FAK protein expression						(0.326)
Negative	14	11 (79)	(0.063)	19	9 (47)	0.013
Positive (scores 2–3)	44	37 (85)	0.641	39	31 (79)	
ASAP1 protein expression						(0.230)
Negative	20	15 (75)	(0.149)	32	19 (59)	0.082
Positive (scores 2–3)	38	33 (87)	0.264	26	21 (81)	
HERG1 protein expression						(0.366)
Negative	5	4 (80)	(0.02)	9	3 (33)	0.006
Positive (scores 2–3)	52	43 (83)	0.882	46	36 (80)	

^#^ Spearman correlation coefficient (in parentheses) with the associated *p* value.

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
