# Peer review of "The Differential Impact of SRC Expression on the Prognosis of Patients with Head and Neck Squamous Cell Carcinoma"

_cancers, 2019, doi:10.3390/cancers11111644_

Round 1

Reviewer 1 Report

This revised manuscript was much improved.

However, detailed data were not presented.

The authors should summarize clinical data including survival information in each case and IHC staining results in each sample in supplementary table to validate their analysis.

Author Response

Comments and Suggestions for Authors

This revised manuscript was much improved.

Response: We thank the reviewer for the positive review report and comments.

However, detailed data were not presented.

Point 1: The authors should summarize clinical data including survival information in each case and IHC staining results in each sample in supplementary table to validate their analysis.

Response 1: Following the reviewer’s recommendation, a new supplementary Table has now been provided including clinical and follow-up data from each patient and the IHC expression analysis of SRC and SRC-related proteins for a total of 116 patients with HNSCC (58 in the larynx and 58 in the pharynx).

We noticed that in the previous version of our manuscript, data on Figure 4 (for CTTN, FAK and ASAP1) was not adequately filtered and plotted for the 116 HNSCC patients precisely studied for SRC IHC, as CTTN, FAK and ASAP1 were analyzed in 12 additional HNSCC patients beyond the present study. To avoid confusion, a new Figure 4 is now provided that perfectly matches data on Supplementary Table S1 for the 116 HNSCC patients.

Reviewer 2 Report

HNSCC is one of the cancers that has a low five-year survival rate and that the known therapies do not increase survival significantly. Therefore, the prediction of HNSCC prognosis would be very important. In this respect, this manuscript is very interesting. Experimental methods and results are well-organized. The number of samples used in the experiment would be sufficient to support the results.

Author Response

Comments and Suggestions for Authors

HNSCC is one of the cancers that has a low five-year survival rate and that the known therapies do not increase survival significantly. Therefore, the prediction of HNSCC prognosis would be very important. In this respect, this manuscript is very interesting. Experimental methods and results are well-organized. The number of samples used in the experiment would be sufficient to support the results.

Response: We thank the reviewer for considering that this is a relevant work and that data are adequately organized and presented.

Reviewer 3 Report

This is an important study about the SRC expression of the prognosis of patients with head and neck squamous cell carcinoma.

Author Response

Comments and Suggestions for Authors

This is an important study about the SRC expression of the prognosis of patients with head and neck squamous cell carcinoma.

Response: We thank the reviewer for highlighting the interest and importance of our study.

This manuscript is a resubmission of an earlier submission. The following is a list of the peer review reports and author responses from that submission.

Round 1

Reviewer 1 Report

This article demonstrates about the SRC expression relating to the prognosis of patients with head and neck squamous cell carcinoma. SRC-related signaling pathway may be interesting to be further investigated in future.

Reviewer 2 Report

Most of the results of this manuscript are already shown in Ref. 41 (n=93). The specimen numbers are also similar (n=122). The only difference is that SRC-dependent prognosis is more specific when the tumor site is larynx. But, simply this content alone cannot represent the originality of this manuscript. In addition, Jiang et al. showed that higher expression level of TrkB was found in laryngeal cancer specimens and TrkB activates AKT via c-Src, leading to increased proliferation. Also, TrkB induced EMT via increased expression of EMT related transcription factors such as Twist-1 and Twist-2.(Oncotarget. 2017 Oct 9;8(65):108726-108737.) Therefore, it is hard to think that this paper contains sufficiently new contents.

Reviewer 3 Report

SRC activation has been reported in several cancers including head and neck squamous cell carcinoma. Human SRC protein is activated by Y419 phosphorylation, and inactivated by Y530 phosphorylation. It has already been reported that high expression of total and Y419-phosphorylated SRC correlates with poor prognosis in nasopharyngeal carcinoma (Ke, et al. Oncotarget 2016. https://doi.org/10.18632/oncotarget.8634).

In this study, the authors examined only the expression of Y530-dephosphorylated form of SRC protein in HNSCC, but not the expression of total and Y419-phosphorylated SRC. However, to elucidate the contribution of Y530 dephosphorylation in SRC to prognosis, not only Y530-dephosphorylated SRC but also total and Y419-phosphorylated SRC must be examined and these expressions must be compared with one another.